# Influence of Testosterone in Neglected Tropical Diseases: Clinical Aspects in Leprosy and In Vitro Experiments in Leishmaniasis

**DOI:** 10.3390/tropicalmed8070357

**Published:** 2023-07-10

**Authors:** Laís Lima de Oliveira Rekowsky, Daniela Teles de Oliveira, Rodrigo Anselmo Cazzaniga, Lucas Sousa Magalhães, Lenise Franco Albuquerque, Jonnia Maria Sherlock Araujo, Martha Débora Lira Tenório, Tiziane Cotta Machado, Michael W. Lipscomb, Priscila Lima dos Santos, Amelia Ribeiro de Jesus, Márcio Bezerra-Santos, Ricardo Luís Louzada da Silva

**Affiliations:** 1Laboratory of Immunology and Molecular Biology, Federal University of Sergipe, Aracaju 49060676, Brazil; lais.lima.oliveira@gmail.com (L.L.d.O.R.); daniela.oliveira@saude.se.gov.br (D.T.d.O.); cazzaniga.rodrigo@gmail.com (R.A.C.); lucas.smagalhaes@hotmail.com (L.S.M.); lenise1978@hotmail.com (L.F.A.); jonniasherlock@gmail.com (J.M.S.A.); martha_dlt@hotmail.com (M.D.L.T.); tizicotta@ig.com.br (T.C.M.); ameliaribeirodejesus@gmail.com (A.R.d.J.); marciobezerra.ufs@outlook.com (M.B.-S.); qjobio@gmail.com (R.L.L.d.S.); 2Posgraduate Program of Health Science, Federal University of Sergipe, Aracaju 49060676, Brazil; 3Dermatology Division of Medical Hospital, Federal University of Sergipe, Aracaju 49060676, Brazil; 4Department of Pharmacology, Center for Immunology, University of Minnesota, Minneapolis, MN 55455, USA; mwlipscomb.umn@gmail.com; 5Instituto de Investigação em Imunologia (iii), Institutos Nacionais de Ciência e Tecnologia (INCT), CNPq, São Paulo 05403-900, Brazil; 6Health Education Department, Federal University of Sergipe, Lagarto 49400000, Brazil

**Keywords:** leprosy, testosterone, macrophage suppression, leishmania, neuroendocrine system

## Abstract

Neglected tropical diseases encompass a group of chronic and debilitating infectious diseases that primarily affect marginalized populations. Among these diseases, leprosy and leishmaniasis are endemic in numerous countries and can result in severe and disfiguring manifestations. Although there have been reports indicating a higher incidence of leprosy and leishmaniasis in males, the underlying factors contributing to this observation remain unclear. Therefore, the objective of this study was to examine both clinical and experimental evidence regarding the role of testosterone in leprosy and leishmaniasis. A prospective clinical study was conducted to compare the clinical forms of leprosy and assess circulating testosterone levels. Additionally, the impact of testosterone on *Leishmania amazonensis*-infected macrophages was evaluated in vitro. The findings demonstrated that serum testosterone levels were higher in women with leprosy than in the control group, irrespective of the multi- or pauci-bacillary form of the disease. However, no differences in testosterone levels were observed in men when comparing leprosy patients and controls. Interestingly, increasing doses of testosterone in macrophages infected with *L. amazonensis* resulted in a higher proportion of infected cells, decreased CD40 expression on the cell surface, elevated expression of SOCS1, and decreased expression of IRF5. These findings provide biological evidence to support the influence of testosterone on intracellular infections, though the interpretation of clinical evidence remains limited.

## 1. Introduction

Neglected tropical diseases are a group of chronic and debilitating infectious diseases that mainly affect people in tropical and subtropical areas, especially economically and socially vulnerable populations [1]. Each year, over 1.7 billion people worldwide require interventions for at least one of these diseases. Additionally, these diseases are responsible for approximately 200,000 deaths and impose high health and social costs on governments due to disabilities, stigmatization, and social exclusion. Among several neglected diseases, leprosy and leishmaniasis are prevalent in different regions, potentially exposing millions of people to infection [2]. However, the relationship between susceptibility and resistance to the diseases among people living in the same endemic area who present different types of responses, including asymptomatic forms, remains unclear.

Leprosy is a chronic and granulomatous infectious disease caused by *Mycobacterium leprae* and *Mycobacterium lepromatosis* [3]. It leads to skin and peripheral nerve lesions, resulting in physical disability [4]. Once the infection is established, leprosy reactions and neurological disability can cause severe clinical complications with aggravated nerve dysfunction [5]. These characteristics of leprosy have significant socioeconomic and psychological impacts on the affected population. On the other hand, cutaneous and diffuse leishmaniasis are parasitic diseases caused by *Leishmania* spp. parasites, which are kinetoplastid parasitic protozoans, such as *Leishmania (Viannia) amazonensis* specie. These diseases have a higher prevalence in India, Africa, and the Americas [6]. The established diseases lead to ulcerated lesions that can evolve into disseminated or mucocutaneous severe forms, resulting in deformities and sequels in the patients [7]. Both diseases are directly associated with immune response involvement, and the progression of infection can be controlled or enhanced depending on the type of immune response developed.

Understanding the influence of hormonal mechanisms on the clinical presentations of leprosy and leishmaniasis can help clinicians to comprehend the regulation of neuroendocrine and immunologic interactions in various diseases, particularly intracellular infections. Several studies have documented sex-based differences in diseases caused by intracellular pathogens, including leprosy [8], tuberculosis [9], leishmaniasis [10,11], and malaria [12]. In addiation, von Steeg and Klein reviewed differences in severity or prevalence in other infectious diseases caused by influenza A, hepatitis B virus, and *Cryptococcus neoformans* infections, among other examples [13]. These findings were initially attributed to a higher exposure of males to these infectious diseases for a prolonged period, though this explanation is now considered superficial and outdated. Experimental data on paracoccidioidomycosis and leishmaniasis support the hypothesis that testosterone plays a role in the susceptibility of males to intracellular infection [14,15]. In vitro studies have described the effects of testosterone on immune modulation in terms of reducing the mRNA expression of several inflammatory response products, such as tumor necrosis factor alpha (TNF-α), interleukin (IL)-6, CD40, toll-like receptor 4 (TLR4), and cyclooxygenase 2 [16]. Moreover, it increases IL-10 production [17] and reduces natural killer cell activity [18]. However, no studies have described any association between testosterone and its role as a possible immune response evasion mechanism against intracellular parasites.

Therefore, the objective of this study was to generate clinical and experimental evidence to support the hypothesis that a biological role exists for testosterone in explaining sex-based differences in leishmaniasis and leprosy cases. A prospective clinical study was conducted to correlate the circulating testosterone levels (before treatment) with clinical forms and outcomes of leprosy complications. Furthermore, the effect of in vitro testosterone treatment on macrophages infected by *L. amazonensis* was investigated.

## 2. Materials and Methods

### 2.1. Comparative Clinical Data for Testosterone Levels in Patients with Leprosy and Control Participants

A prospective study was conducted, which included 119 patients with leprosy who were diagnosed and treated at University Hospital, the Federal University of Sergipe. Nineteen household contacts (HHC) were recruited as the controls. These participants had lived in close contact with the patients for an extended period and showed no clinical signs of the disease (spouses and housekeepers of adopted children).

All patients underwent a comprehensive dermatological examination for clinical assessment. Slit-skin smears were also collected for bacteriological examination. Cutaneous biopsies were histo-pathologically analyzed to classify the patients, according to Ridley and Jopling classification, into indeterminate (IL), tuberculoid (TT), borderline (BL), and lepromatous leprosy patients (LL) [19]. The patients were initiated on multidrug therapy (MDT) based on the operational classification of the World Health Organization (WHO), which categorizes the disease in pauci- (PB) or multi-bacillary (MB) terms [20]. Additionally, leprosy type 1 and 2 reactions were diagnosed in patients during the follow-up period. Physical disability was evaluated based on the standard examinations of the WHO and the Brazilian Ministry of Health, which ranked patients into three degrees (degrees 0, 1, and 2). Patients were assessed at the time of diagnosis, after the initiation of MDT, and at each monthly visit throughout treatment. They were further monitored every 3 months for a duration of 2 years following the completion of treatment to identify the potential occurrence of physical disabilities or reaction episodes.

### 2.2. Blood Collection and Hormone Assays

Blood samples were collected from patients with leprosy at the time of diagnosis, as well as from household contacts. Serum testosterone levels were measured using commercially available automated analyzer immunoassays in a certified laboratory. The reference range for normal testosterone levels by age, which was provided by the kit manufacturer (Abbott Architect Plus i2000, Abbott Park, IL, USA), was used to determine whether the measured levels were within the normal range.

### 2.3. In Vitro Effect of Testosterone Treatment for Infection via LEISHMANIA-GFP in Macrophages

Adult healthy male donors (n = 11) were included in this assay. Blood collection and cell isolation/differentiation were performed based on our previously published protocol [21]. During macrophage differentiation, the culture medium was supplemented with testosterone (1.0 mg/mL, testosterone solution, T-037) (Cerilliant, Round Rock, TX, USA) at fractional doses ranging from 0 to 160 nM. Testosterone-supplemented culture medium was replaced on days 2 and 5 of the differentiation culture. Charcoal-stripped fetal bovine serum (FBS) was used to eliminate any interference from testosterone that could be present in regular FBS. The use of testosterone was based on a previously published protocol [22].

To infect the macrophages, *L. amazonensis* (MHOM/BR/73M2269) [23], which constitutively expresses GFP (Leishmania-GFP), was utilized. Macrophages were infected with stationary-phase promastigotes at a multiplicity of infection of 20:1, and extracellular parasites were removed via washing after 2 h. After 18 h of infection, the cells were harvested from 24-well plates using cold phosphate buffered saline.

The analysis of Leishmania-GFP intracellular infection was performed by measuring the percentage of GFP+ cells, which were considered an indicator of the proportion of infected cells, and the mean fluorescent intensity (MFI) of GFP, which was considered a representation of the number of parasites inside the cells. Additional markers were evaluated in infected macrophages following the previously published protocol [24]. Briefly, after washing, cells were stained with antibodies for CD40 and CD163. Subsequently, the cells were fixed with 4% paraformaldehyde, permeabilized with PermWash (BD Biosciences, San Jose, CA, USA), and stained with IL-4, TNF-α, IL-12, and IL-10 antibodies. Data acquisition was performed using FACS CANTO II (BD Biosciences, San Jose, CA, USA), and data were analyzed using FlowJo v10.0 software (FlowJo LLC, Ashland, OR, USA). Surface marker expression and intracellular cytokine analysis data were represented using the iMFI measure, which was calculated by multiplying the relative frequency (percent positive) of cells expressing a specific cytokine/surface marker with the MFI of that population [25].

### 2.4. Gene Expression Profiling via Quantitative Polymerase Chain Reaction (qPCR)

For mRNA expression analysis, TRIzol™ (Thermo Fisher, Carlsbad, CA, USA) reagent was added to macrophages, and RNA extraction was performed following the manufacturer’s instructions. The extracted RNA was then reverse-transcribed into complementary DNA using the High-Capacity cDNA Reverse Transcription Kit (Thermo Fisher, Carlsbad, CA, USA). Gene expression analysis, specifically quantitative PCR (qPCR), was conducted using TaqMan assays (Thermo Fisher, Carlsbad, CA, USA) that targeted genes of interest. The following probes were utilized: suppressor of cytokine signaling (SOCS) 1, SOCS3, interferon regulatory factor (IRF) 4, and IRF5. An ABI7500 FAST real-time PCR system (Applied Biosystems™, Thermo Fisher, Carlsbad, CA, USA) was used to analyze the samples. The expression levels of the target transcripts were normalized to suit the housekeeping gene GAPDH.

### 2.5. Statistical Analysis

D’Agostino’s and Pearson’s tests were utilized to assess whether the data exhibited a normal distribution. The percentages, means, and standard deviations of the groups were calculated. Hormone concentrations were compared across different subgroups based on the clinical or operational forms of leprosy using Student’s t-test. Demographic and clinical variables between the patient and control groups were compared using either the Kruskal–Wallis or Mann–Whitney test. Fisher’s exact test was employed to calculate the relative risk and confidence interval for categorical data. The Wilcoxon paired test was used to compare gene expression, surface phenotype/cytokine profiles, and in vitro infection analyses. Statistical significance was considered to be achieved when *p* < 0.05. All analyses were conducted using GraphPad Prism version 9.0 (GraphPad Software, Boston, MA, USA).

## 3. Results

### 3.1. Demographic and Clinical Data of Patients with Leprosy Included in the Hormone Study

Table 1 presents the demographic and clinical data of the patients enrolled in the clinical study. Patients with MB leprosy had a higher mean age (mean ± standard deviation; 48.7 ± 17.1 years; *p* = 0.01) than patients with PB leprosy (43.4 ± 18.29 years) and HHCs (36.6 ± 8.45 years; Table 1). The proportion of men with MB (54%) was significantly higher than that of men with PB (46%). Furthermore, the occurrence of leprosy reaction episodes was significantly higher in MB patients (43%) than in PB cases (12%; *p* = 0.01), while physical disabilities were more common in MB patients (77%) than in PB patients (21%; *p* < 0.0001).

### 3.2. Association of Higher Serum Testosterone Levels with the Occurrence and Severity of Leprosy

Circulating testosterone levels were assessed in patients with leprosy, categorized by their operational (PB and MB) and clinical forms, and compared to HHCs. In males, no significant difference in testosterone levels was observed between those with MB and PB leprosy (Figure 1A). Similarly, no significant differences were found when comparing testosterone levels in males based on the Ridley and Jopling clinical forms (Figure 1C).

Interestingly, although testosterone levels were considerably lower in females than in males, females with either PB (*p* = 0.02) or MB leprosy (*p* = 0.03) exhibited higher testosterone levels than HHCs (Figure 1B). When examining females, according to Ridley and Jopling clinical forms, no significant differences in testosterone levels were noted (Figure 1D). However, when comparing testosterone levels and the presence of a leprosy reaction prior to treatment, lower levels of testosterone were detected in the group of females with a leprosy reaction (*p* = 0.02) than in those without it (Figure 1F).

### 3.3. Role of Testosterone in Decreasing In Vitro Macrophage Function in an Intracellular Infection Model

Testosterone treatment demonstrated a dose-dependent effect, as it increased the percentage of infected cells above 80 Nm (*p* = 0.0313) (Figure 2A). However, no significant differences were observed in MFI of GFP, indicating that the number of parasites inside the infected cells remained unchanged (Figure 2B).

Following infection, testosterone-treated macrophages exhibited a gradual decrease in CD40 expression based on comparisons of 0, 80, and 160 nM groups (*p* = 0.04) (Figure 3C). Although there was a trend of increase in CD163 and IL-10 levels, the differences were not statistically significant (Figure 2D,E). No significant differences were observed between IL-4 and TNF-α production (Figure 2F–H). While investigating the impact of testosterone treatment on macrophages in vitro, an upregulation of SOCS1 gene expression (*p* = 0.04) was observed, but no upregulation of SOCS3 was observed (Figure 3A,C). Additionally, there was a reduction in IRF5 mRNA expression (*p* = 0.05), but no significant change in IRF4 expression was observed (Figure 3B,D).

## 4. Discussion

Considering the numerous reported evasion mechanisms that favor parasite survival, it is plausible that micro-organisms utilize hormone pathways to evade the immune response. In a study by Singh et al. (2015) [26], lower levels of testosterone were detected in patients with leprosy than in healthy controls, though the study did not establish a correlation between hormone levels and the MB form of the disease [26].

In the present study, we observed a trend of increased serum testosterone levels in men with MB leprosy compared to those with PB leprosy, although the differences were not statistically significant. Conversely, in females, both the PB and MB forms of leprosy exhibited higher serum testosterone levels than the control group, indicating an association between the hormone and disease outcome. Furthermore, leprosy reactions, which are known for their pathological and exacerbated inflammatory processes, according to the literature [8], were found to occur more frequently in females with higher testosterone levels. This finding supports the immunoregulatory role of testosterone in protecting patients from excessive inflammation.

Plasma hormone levels have been associated with cytokine profile imbalances that influence the outcomes of various infectious diseases, including leprosy and leishmaniasis [27,28,29]. Experimental data further support the existence of sex-based differences in the outcomes of infectious disease [30]. Furthermore, data showing the suppressive effect of testosterone on immune cells, such as dendritic studies, have reported the suppressive effect of testosterone on immune cells, such as dendritic cells and macrophages [22]. Based on this hypothesis, our results demonstrate a dose-dependent effect of testosterone on the suppression of macrophage microbicidal activity during *Leishmania amazonensis* infection, as well as induction of SOCS 1 expression and downregulation of IRF5. These findings are consistent with previous studies and provide evidence that the testosterone pathway may serve as an evasion mechanism that is utilized by intracellular pathogens to evade the immune response [31].

Disease severity is influenced by the proper balance of various pathways that activate microbicidal functions to control the parasite. The suppressive activity of testosterone in macrophages has been linked to reduced expression of TLR4 [32]. In our study, we observed a significant decrease in the expression of CD40, which is an important costimulatory molecule associated with the inflammatory profile in response to IFN-γ, in testosterone-treated infected cells [33]. This result suggests that testosterone renders macrophages less susceptible to lymphocyte activation toward an effector profile. No differences were observed in the expression of other surface markers or cytokines. Furthermore, our findings demonstrated an upregulation of SOCS1, but not of SOCS3, as well as downregulation of IRF5, but not of IRF4, expression in testosterone-treated macrophages. These observations suggest that these molecular changes may contribute to the suppressive effect of testosterone on macrophages during *Leishmania* intracellular infection. SOCS and IRF-signaling proteins have been extensively associated in past studies with either a resolutive or worsened immune response in infectious diseases [34,35]. However, further investigations that use other components of these immune cell activation/suppression cascades are warranted to gain a more comprehensive understanding of the underlying mechanisms.

In summary, our findings provide evidence to support the hypothesis that interactions between *M. leprae* and *L. amazonensis* infections, immunological function, and the endocrine system play crucial roles in the progression of intracellular infections. The integration of clinical data and in vitro assays substantiates the immunoregulatory effect of testosterone on human macrophages and highlights the significance of this hormone in host–pathogen interactions. Moreover, our study emphasizes the importance of closely monitoring testosterone administration in general clinical care.

## Figures and Tables

**Figure 1 tropicalmed-08-00357-f001:**
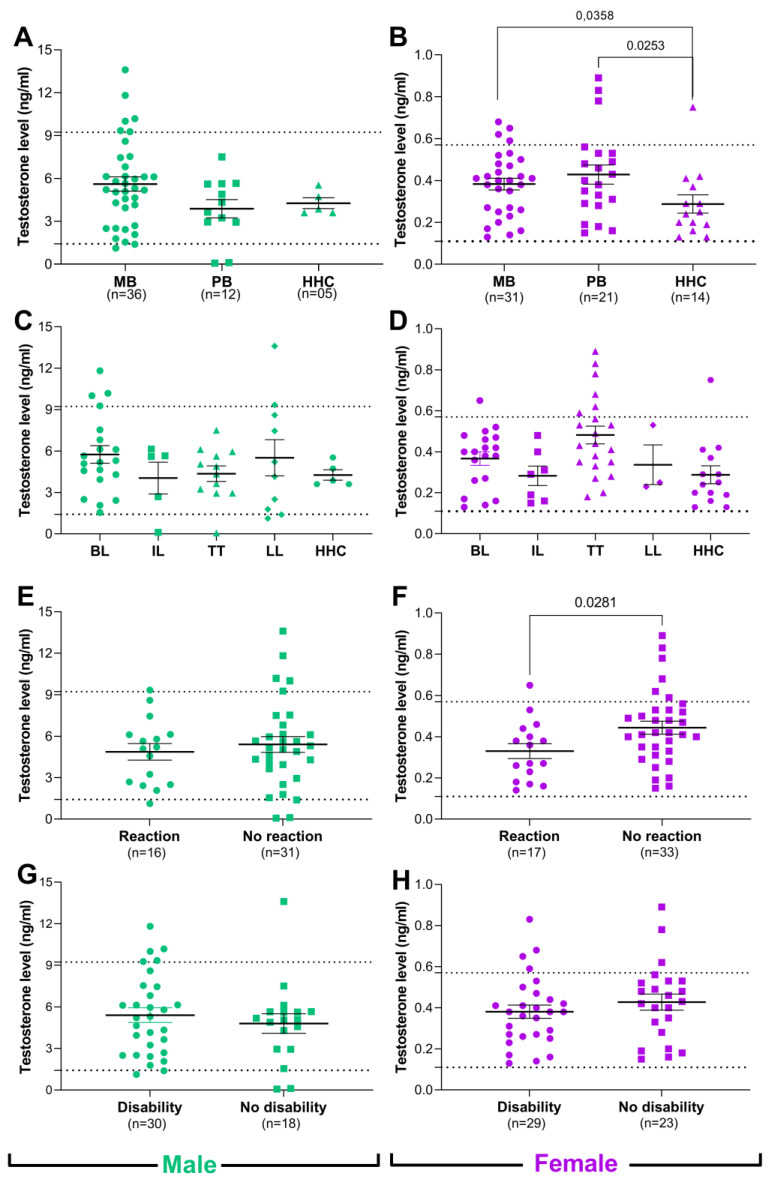
Testosterone levels in all patients with leprosy according to their operational and clinical forms. Testosterone levels were measured in male patients with PB and MB and HHCs (**A**). Male patients with leprosy were classified based on their clinical forms determined via histopathology (**B**). Testosterone levels were measured in female patients with PB and MB and HHCs (**C**), as well as the leprosy classification based on their clinical forms determined via histopathology (**D**). Testosterone levels were compared in female (**E**) and male (**F**) patients with or without leprosy reaction. The presence of disability was also evaluated in male (**G**) and female (**H**) patients. Sera from the patients were collected, and the hormone concentrations were determined in a certificated laboratory via the standard methods. Dotted lines indicate the assumed normal limits of testosterone levels in the population. Unpaired t-test was performed.

**Figure 2 tropicalmed-08-00357-f002:**
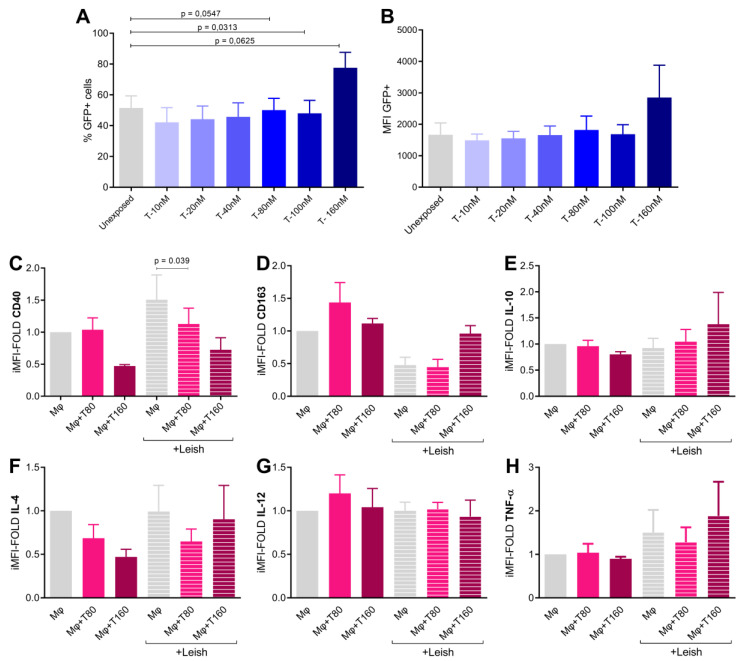
Effect of testosterone treatment on macrophages in an in vitro intracellular infection model. Mononuclear cells were treated with testosterone solution in fractional doses (T-0 to T-160 nM) and infected with Leishmania-GFP (MOI = 20:1, 18 h after infection) prior to analysis via flow cytometry (n = 11 experiments). Data express percentages of infected cells (GFP+ cells) (**A**) and GFP MFI analysis (**B**). Cytokines and macrophage markers were also evaluated (**C**–**H**). Bars represent uninfected and infected cells treated with testosterone (0 nM, 80 nM and 160 nM, respectively). CD40 and CD163 cell surface expression (n = 11 experiments) are depicted in (**C**,**D**), respectively. Cytokine profiles of IL-10 (**E**), IL-4 (**F**), IL-12 (**G**), and TNF-α (**H**) (n = 6 experiments). Treated groups were compared to the control group (without testosterone treatment) via Wilcoxon paired tests.

**Figure 3 tropicalmed-08-00357-f003:**
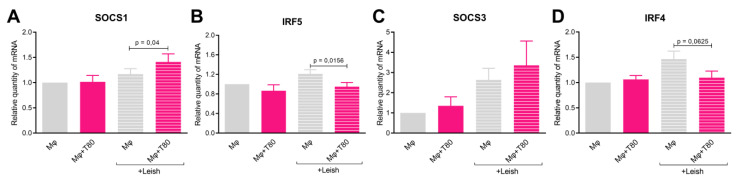
Testosterone effects on gene expression in infected macrophages. PBMCs were collected from healthy male donors, and the adherent cells were cultured for five days in RPMI plus 10% charcoal-FBS. Cells were treated with testosterone solution in fractional doses (T-0 to T-160 nM) and infected with Leishmania-GFP (MOI = 20:1, 18 h after infection) prior to qPCR analysis, and the treatment (80 nM) and control groups (0 nM) were compared. (**A**) SOCS1, (**B**) IRF5, (**C**) SOCS3, and (**D**) IRF4 are depicted. Groups were compared via Wilcoxon paired tests (n = 7 experiments).

**Table 1 tropicalmed-08-00357-t001:** Demographic and clinical data of leprosy patients included in this clinical study.

Variables	MB (n = 67)	PB (n = 33)	HHC (n = 19)	OR [CI 95%]	*p*-Value ^†^
Age					
Variation	12–82	05–79	19–53	–	*p* = 0.01 ^‡^
Mean ± SD	48.73 ± 17.18	43.48 ± 18.29	36.6 ± 8.45		
Sex					
Male n (%)	36 (54%)	12 (36%)	05 (27%)	2.03 [0.88 to 4.58]	*p* = 0.13
Female n (%)	31 (46%)	21 (64%)	14 (73%)		
Leprosy Reaction					
n (%)	29 (43%)	04 (12%)	NA	5.53 [1.86 to 15.72]	*p* = 0.01
Physical Disability					
n (%)	52 (77%)	07 (21%)	NA	12.88 [4.51 to 31.87]	*p* < 0.0001

† Fisher’s exact test. ‡ ANOVA with Tukey’s post-test. MB, multibacillary; PB, paucibacillary; HHC, household health controls; OR, odds ratio.

## Data Availability

Additional data can be obtained from the corresponding author upon reasonable request.

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
