# Peer review of "Influence of Testosterone in Neglected Tropical Diseases: Clinical Aspects in Leprosy and In Vitro Experiments in Leishmaniasis"

_tropicalmed, 2023, doi:10.3390/tropicalmed8070357_

Round 1

Reviewer 1 Report

Good work

Major corrections to the language are essential.

Good work

Major corrections to the language are essential.

Author Response

Dear reviewer, we updated the manuscript improving language. Thank you.

Reviewer 2 Report

The present research work is well written, easy to read and follow, and presents extremely interesting data about the relationship between testosterone and the possible influence on clinical and experimental aspects in infectious leprosy and leishmaniasis. The issue raised is of great interest because there is currently a gap in this issue. 

The methodology is well thought out and supported by robust statistical analysis. The results and conclusions are very good, duly supported, and very well illustrated for interpretation by the public. 

Some minor modifications to some errors found in the text are suggested. 

>The use of italics in microbial species (Mycobacterium leprae and Leishmania amazonensis) in lines 51 and 52 and in the text in general

>Use of italics for Latin words (in vitro) on line 83 and et al on line 223 and throughout the text

>Lines 56 and 57 reveal a conceptual and/or drafting error. Leishmaniasis is a parasitic disease caused by different species of Leishmania. The phrase "... cutaneous and diffuse leishmaniasis is a parasitic disease caused by Leishmania (Viannia) amazonensis in Americas, among other involved species..." is not precise because it tries to indicate that this is the main species in all the Americas which is not correct. It is suggested to improve the wording

>Lines 66 and 67 suggest mentioning intracellular diseases different from those studied in this article to give greater support to the research question

Author Response

Dear Reviewer, we appreciate the feedback and suggestions.

Points to modifications:

  1. Lines 51 and 52, we are sorry for these mistakes. All essential italic formatting was added.
  2. Lines 56 and 57: Yes, previously there are some manuscripts drafting. We improved the language in the manuscript. Thank you
  3.  Lines 66 and 67: Thank you for the suggestion. In the revised version new examples were added.

Reviewer 3 Report

The manuscript contains experimental results that may lead to the control of leishmaniasis and its concomitant association with leprosy although may be considered preliminary results, the science is solid and the methods and interpretations are sound; however, the manuscript is in need of english editing and clarification of some passages the are confusing or misspelled; further details are included. 

Abstract.

Line 25 …”disfiguring forms”… incomplete sentence; should add …of both diseases.

Line 29 should make clear if the infected macrophages were in vitro evaluated. 

Line 32 … There IS no difference… incorrect conjugation, subjects of sentence in plural.

Introduction.

Some english editing is required, the redaction should be improved for a better understanding.

Lines 41-50. Some sentences are incomplete or misspelled, a careful english editing by a profesional dedicated service should be applied.

Line 57. When Leishmania genus is mentioned without species should be written Leismania sp and should include a basic taxonomical description like “kinetoplastid parasitic protozoan”

Results

Lines 120-124. Is there any citation for L. amazonensis MHOM/BR/73M2269 strain origin?. 

Lines 159-165. Acronyms used in this paragraph 1 should be described below of table 1, those acronyms described at the M&M section are different than those used in this paragraph.

The manuscript requires english editing and clarification of some passages the are confusing or misspelled; further details are included. 

Author Response

Dear Reviewer,

We appreciate the suggestions. The language was extensively revised.

Points for review:

  1. Abstract: the abstract was updated and these sentences was changed.
  2. Introduction: Various revisions were made in the writing.
  3. Lines 41-50: We are sorry about this, the text was modified.
  4. Line 57: Thank you for the suggestion, We updated in manuscript.
  5. Results: Sorry for this. We added the reference for the stain.
  6. Lines 159-165: We are sorry about for this mistake, we changed acronymous and added description in the table.

Reviewer 4 Report

The study shows the association between testosterone  and the evasion of immune response to intracellular pathogens; M. leprae & L. amazonensis. The authors used a prospective study to measure the plasma testosterone in different clinical presentations of leprosy in males and females. For leishmania, they used Invitro testosterone-treated macrophages to show the immunomodulatory effect of testosterone. 

minor comments:

1- In material and methods:

-I think the authors need to mention the reference for testosterone treatment of macrophages.

- Staining for surface markers before FACS: It is impossible to fix the cells the use of surface staining CD40 and CD162. The protocol for staining is surface staining of cells then fixation and permeablization for the intracellular staining. Please fix the text.

- Plaese add Wilcoxon test to your statistical analysis part. 

Author Response

Dear Reviewer,

we appreciate the suggestions.

Minor comments:

  1. Material and methods: we included the reference that we used in our assays. Sorry for this.
  2. Staining markers: This was an error in the manuscript  writing. Now was updated.
  3. Wilcoxon test: The test was added in the subsection.

Round 2

Reviewer 3 Report

The revised manuscript is now acceptable for publication